# Impact of SARS-CoV-2 Positivity on Delivery Outcomes for Pregnant Women between 2020 and 2021: A Single-Center Population-Based Analysis

**DOI:** 10.3390/jcm12247709

**Published:** 2023-12-15

**Authors:** Raffaele Palladino, Federica Balsamo, Michelangelo Mercogliano, Michele Sorrentino, Marco Monzani, Rosanna Egidio, Antonella Piscitelli, Anna Borrelli, Giuseppe Bifulco, Maria Triassi

**Affiliations:** 1Department of Public Health, University “Federico II” of Naples, 80131 Naples, Italymichelangelo.mercogliano@gmail.com (M.M.); sorrentinoemme@gmail.com (M.S.); marco.monzani@unina.it (M.M.); giuseppe.bifulco@unina.it (G.B.);; 2Department of Primary Care and Public Health, Imperial College School of Public Health, London SW7 2BX, UK; 3Interdepartmental Research Center in Healthcare Management and Innovation in Healthcare (CIRMIS), University “Federico II” of Naples, 80131 Naples, Italy; 4Clinical Directorate, Academic Hospital “Federico II” of Naples, 80131 Naples, Italy; 5Azienda Ospedaliera di Rilievo Nazionale (AORN) Dei Colli, Vincenzo Monaldi Hospital, 80122 Naples, Italy

**Keywords:** pregnancy, COVID-19, delivery, birth, vaccination, SARS-CoV-2

## Abstract

Despite the existing body of evidence, there is still limited knowledge about the impact of SARS-CoV-2 positivity on delivery outcomes. We aimed to assess the impact of SARS-CoV-2 infection in women who gave birth at the University Hospital “Federico II” of Naples, Italy, between 2020 and 2021. We conducted a retrospective single-center population-based observational study to assess the differences in the caesarean section and preterm labor rates and the length of stay between women who tested positive for SARS-CoV-2 and those who tested negative at the time of labor. We further stratified the analyses considering the time period, dividing them into three-month intervals, and changes in SARS-CoV-2 as the most prevalent variant. The study included 5236 women with 353 positive cases. After vaccination availability, only 4% had undergone a complete vaccination cycle. The Obstetric Comorbidity Index was higher than 0 in 41% of the sample. When compared with negative women, positive ones had 80% increased odds of caesarean section, and it was confirmed by adjusting for the SARS-CoV-2 variant. No significant differences were found in preterm birth risks. The length of stay was 11% higher in positive cases but was not significant after adjusting for the SARS-CoV-2 variant. When considering only positive women in the seventh study period (July–September 2021), they had a 61% decrease in the odds of receiving a caesarean section compared to the fourth (October–December 2020). Guidelines should be implemented to improve the safety and efficiency of the delivery process, considering the transition of SARS-CoV-2 from pandemic to endemic. Furthermore, these guidelines should aim to improve the management of airborne infections in pregnant women.

## 1. Introduction

The COVID-19 pandemic started in January 2020 and has served as a benchmark for health systems, particularly for emergency departments [1]. The SARS-CoV-2 virus can cause respiratory infections, including severe pneumonia, and in a small percentage of cases, it can lead to death [2,3,4]. Pregnancy can influence the response to SARS-CoV-2 due to the physiological changes that affect the cardiorespiratory and immune systems [5].

Early findings indicated that pregnancy was associated with a reduced risk of manifesting low or mild symptomatic COVID-19 [6]. However, pregnant individuals faced a higher risk of experiencing severe COVID-19, leading to a higher rate of admission to the ICU, the need for supplemental oxygen and ventilation, and mortality [7,8], especially in cases where there were underlying medical conditions such as a higher BMI, diabetes, or cardiopulmonary health issues [9,10].

Despite strong recommendations for vaccination, which was provided free of charge [11] for pregnant and breastfeeding women, a significant level of vaccine hesitancy was reported [12]. The available evidence showed that pregnant women tended to have a lower vaccination uptake as well as low levels of acceptancy and willingness to be vaccinated [13].

There is limited knowledge regarding the impact of SARS-CoV-2 positivity on delivery-related outcomes. The most common pregnancy-related complications associated with SARS-CoV-2 infection appear to be premature rupture of membranes (PROM) (5%), fetal distress (14%), and postpartum fever (8%) [14,15], with a higher risk of requiring caesarean section and experiencing preterm delivery compared to uninfected pregnant women [16]. The evidence suggests that, despite what was recommended in clinical guidelines, the positivity to SARS-CoV-2 might lead the caregiver to prefer a caesarean delivery [17]. Some studies have reported a reduced risk of preterm birth during the pandemic period, as compared to the pre-pandemic years [18,19]. However, the current evidence relies on minimal data and relatively short follow-ups [14,15,16].

The Department of Gynecology at the University Hospital “Federico II” of Naples, situated in the Campania region, one of Italy’s largest and most densely populated regions, served as the regional tertiary referral center for pregnant women who tested positive for SARS-CoV-2 during the study period. Therefore, by analyzing real-world data on the pregnant women admitted to this academic hospital, we aim to contribute to the assessment of the impact of SARS-CoV-2 positivity on the health outcomes of pregnant women at the time of delivery and subsequently improve the management of SARS-CoV-2 positive patients. Consequently, this study was conducted using anonymized data from pregnant women who gave birth at the University Hospital “Federico II” of Naples between January 2020 and December 2021 to assess the impact of SARS-CoV-2 positivity on pregnancy at the time of delivery.

## 2. Methods

### 2.1. Study Design and Settings

We conducted a retrospective population-based observational study involving all the women who gave birth at the University Hospital “Federico II” in Naples between January 2020 and December 2021. To assess the differences in the caesarean section rates, preterm labor rates, and length of stay between women who tested positive for SARS-CoV-2 and those who tested negative at the time of labor, we employed a time series analysis. We also conducted further stratified analyses, considering the study periods divided into three-month intervals and changes in the prevalent SARS-CoV-2 variant. All the records of women who gave birth at the University Hospital “Federico II” in Naples between January 2020 and December 2021 were included in the analysis if they contained all the necessary information.

These analyses were carried out at a single center, as the University Hospital “Federico II” served as the regional hub for the clinical management of SARS-CoV-2 positive pregnant women during the specified period. All the patients provided informed consent authorizing the use of anonymized, routinely collected healthcare data in compliance with the data protection regulation “GDPR EU 2016/679”. The study received ethical approval from the “Federico II University” Ethics Committee with protocol 332/21 and was conducted in accordance with good clinical practice and the Declaration of Helsinki.

### 2.2. Study Variables

The study’s outcomes included preterm birth (defined as delivery at a gestational age < 37 + 0 weeks), the delivery mode (caesarean or vaginal), and the length of hospital stay. The main variable of interest was SARS-CoV-2 positivity, which was determined either within 48 h before hospital admission or at the time of admission through a triage process conducted at the Obstetrical Emergency Unit. The assessment of SARS-CoV-2 positivity was carried out using the polymerase chain reaction (PCR) test for COVID-19. In the instances where a rapid antigen test yielded a positive result during triage at the Obstetrical Emergency Unit, a PCR test was promptly administered to confirm the result. Additionally, information regarding the SARS-CoV-2 vaccination status was obtained exclusively for women who tested positive for SARS-CoV-2. The observation period was further categorized based on the most prevalent SARS-CoV-2 variant at the time of the delivery [20,21,22]. The other variables considered in the study included the age, vaccination status (for positive women), and the Obstetric Comorbidity Index (OB-CMI) [23,24]. We opted for this measure due to its suitability for calculation using hospital discharge records, and it has been previously employed as a proxy for multimorbidity in similar studies [25,26].

### 2.3. Statistical Analysis

Univariate statistical analyses were conducted as needed to describe our study population. To assess the probability of preterm labor and caesarean section between SARS-CoV-2 positive women and the controls, we employed multivariable logistic regression models. Additionally, we used a multivariable Poisson regression model to evaluate the differences in the length of hospital stay. These models were adjusted for various factors, including SARS-CoV-2 status, age, the Obstetric Comorbidity Index, and the time of the year in which the delivery occurred (divided into three-month intervals, e.g., January–March, April–June, July–September, and October–December).

We also conducted additional analyses within a sub-population, restricting both the study population and statistical analysis to only those who were SARS-CoV-2 positive at the time of labor. Certain specific study periods were excluded from the analyses if either no positive women were admitted or if the number of positive cases was extremely low due to the sample size constraints. Our estimates were presented as odds ratios (ORs), incidence rates (IRRs), and 95% confidence intervals (95% CIs), as appropriate. Significance was determined based on *p*-values lower than 0.05. All the statistical analyses were carried out using Stata MP 17.0.

### 2.4. Sensitivity Analysis

Given the recent literature, which indicated that clinical manifestations of infection by different SARS-CoV-2 variants were characterized by different types and severity of symptoms [2,8], sensitivity analyses were conducted. In these analyses, we adjusted for time periods based on changes in the most prevalent SARS-CoV-2 variant, rather than using three-month study periods. In the cases where the analyses were limited to positive women, we excluded the Omicron period if there were too few recorded cases in that specific period due to the sample size constraints.

## 3. Results

We obtained data for 5236 pregnant women and deliveries that occurred between January 2020 and December 2021. In 2020, there were 2682 births, with 5.6% (149) from women who tested positive for SARS-CoV-2 at the time of labor. In 2021, there were 2554 births, with 8% (204) from women who tested positive for SARS-CoV-2 at the time of labor. The overall average age of the mothers was 32.3 years (SD 5.45), and significant differences were observed between the positive and negative cases, as well as between the two years examined in this study (Table 1).

The patients were categorized based on the predominant SARS-CoV-2 variant at the time of birth. Out of the 353 births from pregnant women who tested positive for SARS-CoV-2 in our study, 21% (75) occurred when the alpha variant was predominant between March 2020 and September 2020 or between March 2021 and June 2021, 52% (183) when the beta variant was prevalent between October 2020 and February 2021, 23% (81) when the delta variant was predominant between July 2021 and November 2021, and 4% (14) during the emergence of the omicron variant in December 2021 (Table 1).

The majority of women had no comorbidities included in the OB-CMI, with 59.2% of women having an OB-CMI score of 0.

In early June 2021, the SARS-CoV-2 vaccination became available to the general population. Among the 100 newborns delivered by mothers who had tested positive when the vaccination was widely accessible, only four were born to mothers who had completed a full vaccination cycle (Table 1).

The women who tested positive for SARS-CoV-2 at the time of labor exhibited 80% increased odds of undergoing a caesarean section (OR 1.80; 95% CI 1.43–2.27) compared to the controls. This analysis was adjusted for age, the OB-CMI, and the trimester of birth, with the study period divided into three-month intervals. No significant differences were observed between the two populations in terms of the risk of preterm birth throughout the study period. However, those who tested positive for SARS-CoV-2 at the time of labor had an 11% higher incidence rate of the length of stay (IRR 1.11; 95% CI 1.004–1.22) (Figure 1).

When we restricted our analyses to individuals who tested positive for SARS-CoV-2 at the time of labor, which was conducted to better assess the change in management of this specific population over time, we found that, compared with the October–December 2020 study period, during the July–September 2021 study period, the odds of a caesarean section decreased by 61% (OR 0.39, 95% CI 0.19–0.79). No differences were observed in the risk of preterm delivery or the incidence rate of the length of stay (Table 2).

### Sensitivity Analysis

When we stratified our analyses based on the change in the most prevalent SARS-CoV-2 variant over time to focus on whether this factor might explain the differences [27,28,29], rather than using three-month study periods, no differences were observed in the risk of preterm birth. Similarly, there were no significant differences in the mean length of stay between the women who tested positive and negative for SARS-CoV-2 at the time of labor, or when we restricted our analyses to only women who tested positive for SARS-CoV-2. However, the differences we found in the odds of a caesarean section remained consistent with what we observed in the main analysis (OR 1.88, 95% CI 1.50–2.37; Figure 1).

In the sub-analysis, where we restricted the analysis to those who tested positive for SARS-CoV-2, no differences were found (Table 2).

## 4. Discussion

In this retrospective single-center population-based observational study, which included data on the entire population of women who gave birth at the University Hospital ‘Federico II’ of Naples, the regional tertiary referral center for pregnant women during the pandemic between January 2020 and December 2021, we discovered that women who tested positive for SARS-CoV-2 at the time of labor were 80% more likely to undergo a caesarean section compared to those who tested negative. This finding remained consistent even when it was adjusted for the most prevalent SARS-CoV-2 variant at the time of delivery. On average, 47% of positive women had a caesarean section. This percentage was significantly higher than both the Italian and European rates, 31% and 26%, respectively, while it was consistent with regional statistics. [30,31]. In Italy, several factors impacted the increased rate of cesarean sections, such as the perception of professional risk [32], low adherence to recommendations [33], and the pursuit of profit in private facilities [34]. In contrast, we did not identify any differences in the risk of preterm birth between the women who tested positive and those who tested negative for SARS-CoV-2. Additionally, we observed an 11% increase in the average length of hospital stay for those who tested positive for SARS-CoV-2. However, this observation did not hold true when we adjusted the analyses for the most prevalent SARS-CoV-2 variant.

When we confined the analyses to only positive women, a remarkable 61% decrease in the odds of undergoing a caesarean section was noted in the seventh (VII) study period (July–September 2021) when compared to the fourth (IV) study period (October–December 2020). No other differences were identified in these analyses. 

Our finding regarding the increased odds of receiving a caesarean section confirmed what was observed worldwide during the pandemic [17]. While vaginal delivery was strongly recommended, even for patients who had tested positive for SARS-CoV-2 [35], various clinical considerations may have influenced this outcome. These factors included the heightened risk of obstetric comorbidities, the prevention of transmission, and the management of COVID-19 itself [36]. Nonetheless, this trend was reported long before the pandemic started [34], and therefore this result needs to be further investigated. Interestingly, when restricting our analysis to women who had tested positive for SARS-CoV-2, the odds of a caesarean section decreased by 61% during the seventh (VII) study period (July–September 2021) when compared to the fourth (IV) study period, which represented a low-prevalence period for SARS-CoV-2 [37]. However, when we adjusted for the most prevalent SARS-CoV-2 variant in the positive-tested patients, no differences were found. This suggested that further studies may be necessary to explore this potential association more comprehensively. 

We found no differences in the risk of preterm birth between negative and positive-tested women. This aligned with the results of a recent meta-analysis of 40 studies that showed no overall association, although it did reveal some variations in specific subgroups [38]. These differences may be attributed to a combination of positive factors, such as improved hygienic conditions, reduced work-related stress, decreased air pollution, and negative factors, such as anxiety. Additional factors contributing to these findings may include delays in care due to lockdown measures, reduced physical activity, and the varying impact of these factors in settings with more stringent and prolonged contingency measures [39,40,41].

The increase in the length of stay observed in positive-tested women was only evident when adjusting for the study period, which was consistent with the previous literature [42]. This finding could be attributed to the increased challenges in managing patients with airborne transmitted viral infections during pregnancy, both in terms of clinical and hygienic management [43,44]. Our results indicated that the management of infected patients during pregnancy could be considered suboptimal considering the consistency of this evidence. However, further studies should be conducted considering a longer study period and more granular information to better assess the quality of management. The experience of pandemics has served as a valuable test of the efficiency of our health systems and has provided an unprecedented wealth of evidence that should be utilized to enhance healthcare processes, including pregnancy outcomes.

## 5. Strength and Limitations

This study provided insights on the delivery outcomes in pregnant women based on their SARS-CoV-2 status within a large academic hospital in Southern Italy. Given that this hospital served as the regional tertiary referral center for pregnant women who tested positive, our data were likely representative of the Southern Italy population. Moreover, we employed a robust study design that not only compared positive pregnant women with negative controls but also examined the outcomes within the cohort of positive women to assess the changes specific to this group. These changes might have been influenced not only by the evolution of the pandemic but also by shifts in public health policies and improvements in the quality of care for patients affected by COVID-19. 

However, several caveats merit discussions. First, as we extracted our data from hospital discharge records, miscoding and misclassification might have occurred. However, it is important to note that this probability should be consistent among both SARS-CoV-2 positive and negative women, which may mitigate the potential bias’s negative impact. Second, we extracted data from a tertiary referral university hospital where the health status and pregnancy conditions of the admitted SARS-CoV-2 negative women might have differed from those of the average pregnant women admitted in different healthcare settings within the region. However, based on our findings, 56% of SARS-CoV-2 negative pregnant women admitted for birth delivery to the University ‘Federico II’ of Naples had an OB-CMI equal to zero, which could be considered a proxy of good health status [24]. Third, the length of stay for mothers might have also been influenced by the health status of the newborns, as discharges of the mother and the newborn often coincide. Fourth, the data retrieved for this study was based on the content of the hospital discharge record, which did not include data on previous SARS-CoV-2 infections. Nonetheless, since very few cases of SARS-CoV-2 were notified during the first study period in the Campania region, this might have mitigated the impact of this possible bias. Fifth, the Campania region is well known for a higher rate of registered caesarean sections [31]. Furthermore, the complexity of cases that are treated in the University Hospital “Federico II” of Naples, a tertiary referral center, is greater than average. Hence, those two factors might have influenced the proportion of pregnant woman undergoing a caesarean section. Nonetheless, this study focusing on a high complexity setting provided information for the management of emergency settings that could be misrepresented within the general setting. Sixth, the data was retrieved from a single center, and this potentially might have affected the generalization of the results. However, during the pandemic, the “Federico II” hospital was the referral center for SARS-CoV-2 positive pregnant women. Additionally, considering the total amount of deliveries analyzed (6% of regional total), the relatively larger sample of positive women admitted to the study might have helped to mitigate this potential limitation. Therefore, further analyses should also focus on the newborns’ outcomes, which could help explain the variability in the delivery outcomes, including preterm birth and length of stay.

## 6. Conclusions

The present study included data on 5236 pregnant women, 353 of whom tested positive for SARS-CoV-2. These women were admitted to the University Hospital “Federico II” of Naples, the largest academic hospital in Southern Italy and the regional birthing referral center, for childbirth between January 2020 and December 2021. It is worth noting that we observed a very low vaccination coverage among positive women, with only 4% having received the vaccine.

Our findings provided data from a high complexity setting that might help to better assess the impact of SARS-CoV-2 in managing patients during delivery and improve the generalizability of existing evidence. In particular, they revealed that SARS-CoV-2 positivity was associated with increased odds of undergoing a caesarean section, while no differences were found in the risk of preterm birth between positive and negative pregnant women. However, when adjusting for three-month intervals, we observed an increase in the length of stay for those who tested positive for SARS-CoV-2.

In light of these results, we recommend the implementation of guidelines to enhance the safety and efficiency of the delivery process, especially considering the gradual transition of COVID-19 from a pandemic to an endemic situation. These guidelines should also focus on improving the management of airborne infections in pregnant women. Further studies focusing on the comparison of delivery outcomes between settings that differ in clinical case complexity and activity volume might help to better assess the impact of airborne communicable diseases during delivery as well as identify the drivers of improved management. In conclusion, the insights provided from this paper might concur to improve the knowledge related to SARS-CoV-2 positive pregnant women.

## Figures and Tables

**Figure 1 jcm-12-07709-f001:**
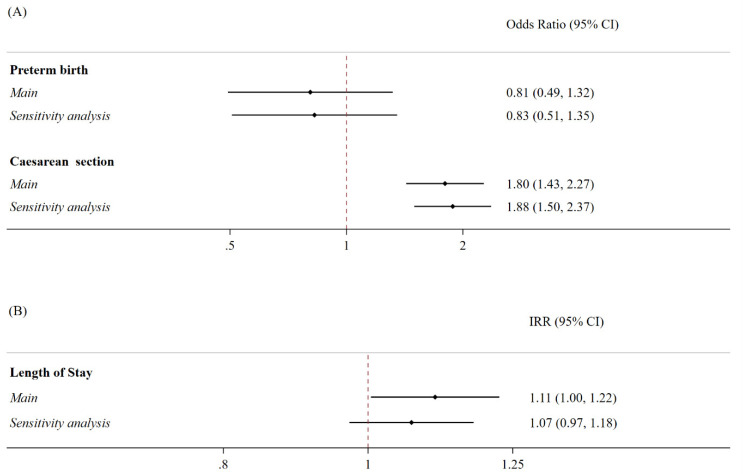
Association between SARS-CoV-2 positivity and the risk of preterm birth, caesarean section deliveries, and percentage increase in the hospital length of stay for delivery. Notes: (**A**) The binary outcomes were modeled using multivariable logistic regression models, controlled for age at the time of delivery, caesarean section (if not the outcome), and the Obstetric Comorbidity Index. For the main analyses, the models were further adjusted for three-month study periods, whilst the sensitivity analyses for time were divided according to changes in the most prevalent SARS-CoV-2 variant, rather than the three-month study period. The analyses were presented as odds ratios (ORs) and 95% confidence intervals (95% CI). C-section: caesarean section. (**B**) The outcomes were modeled by employing multivariable Poisson regression models, controlled for age at the time of delivery, caesarean section, and Obstetric Comorbidity Index. For the main analyses, the models were further adjusted for three-month study periods, while in the sensitivity analyses, time was divided according to changes in the most prevalent SARS-CoV-2 variant, rather than three-month study periods. The analyses were presented as incidence rates (IRRs) and 95% confidence intervals (95% CI).

**Table 1 jcm-12-07709-t001:** Descriptive characteristics of the study population stratified by year and SARS-CoV-2 status.

	2020			2021			Total		
	SARS-CoV-2 +	SARS-CoV-2 −	*p*-Value	SARS-CoV-2 +	SARS-CoV-2 −	*p*-Value	SARS-CoV-2 +	SARS-CoV-2 −	*p*-Value
N	149	2533		204	2350		353	4883	
Average age of mothers (mean, SD)	30.1 (SD 5.3)	32.3 (SD 5.3)	<0.01	30.5 (SD 5.8)	32.7 (SD 5.5)	<0.01	30.3 (SD 5.6)	32.5 (SD 5.4)	<0.01
Prevalent SARS-CoV-2 variant									
*Before the outbreak*—Jan ‘20 to Feb ’20	-	446	<0.01	-	-		-	446	<0.01
*Alpha*—March to Sept ‘20 + March to June ‘21	2 (1%)	1444		73 (36%)	717	0.241	75 (21%)	2161	
*Beta*—Oct ‘20 to Feb ‘21	147 (99%)	643		36 (18%)	372		183 (52%)	1015	
*Delta*—July to Nov ‘21	0	0		81 (40%)	1103		81 (23%)	1103	
*Omicron*—Dec ‘21	0	0		14 (7%)	158		14 (4%)	158	
Vaccine availability	-	-		100	-		100	-	
Complete vaccine cycle	-	-		4	-		4	-	
Obstetric Comorbidity Index									
0	109	1456	0.008	149	1284	0.000	258	2740	<0.01
1–2	38	892		46	924		84	1816	
3–4	0	75		3	70		3	145	
5–6	1	74		6	52		7	126	
7–8	1	28		0	16		1	44	
9–10	0	3		0	2		0	5	
>10	0	5		0	2		0	7	
Delivery mode									
Vaginal	55	1363	0.000	93	1232	0.061	148	2595	<0.01
Caesarean section	94	1170		111	1118		205	2288	
Pregnancy									
Full term	144	2404	0.344	190	2185	0.932	334	4589	0.625
Preterm birth	5	129		14	2185		19	294	
Length of stay (median, IQR)	5 (4 to 8)	5 (4 to 8)	0.834	5 (4 to 8)	4 (4 to 6)	0.000	5 (4 to 8)	5 (4 to 7)	0.014

**Notes**: The descriptive statistics included *t*-tests, analyses of variance, and Chi-square and Wilcoxon rank-sum tests as appropriate. IQR = interquartile range, SD = standard deviation.

**Table 2 jcm-12-07709-t002:** Differences in the risk of preterm birth, caesarean section deliveries, and differences in the hospital length of stay for delivery in SARS-CoV-2 positive women during the study period of birth or from the SARS-CoV-2 prevalent variant.

	Preterm BirTH			Caesarean Section			Length of Stay		
Main Analysis	OR	95% CI		OR	95% CI		IRR	95% CI	
IV study period (ref) (October–December 2020)									
V study period (January–March 2021)	2.10	0.53	8.25	0.60	0.32	1.12	1.18	0.93	1.50
VI study period (April–June 2021)	1.94	0.42	8.90	0.79	0.39	1.57	1.21	0.98	1.48
VII study period (July–September 2021)	3.70	0.99	13.79	**0.39**	**0.19**	**0.79**	1.00	0.77	1.29
VIII study period (October–December 2021)	0.57	0.06	5.09	1.00	0.52	1.93	1.01	0.78	1.32
**Sensitive Analysis**	OR	95% CI		OR	95% CI		IRR	95% CI	
Alpha (ref)									
Beta	0.50	0.15	1.68	0.95	0.54	1.66	0.83	0.66	1.03
Delta	1.07	0.31	3.72	0.67	0.35	1.28	0.80	0.61	1.06
Omicron	-	-	-	1.57	0.48	5.15	0.77	0.48	1.23

**Notes:** The outcomes were modeled by employing multivariable Poisson regression models or multivariate logistic regression models for the binary outcomes, controlled for the three-month study period of delivery, age at time of delivery, caesarean section (if not the outcome), and Obstetric Comorbidity Index. For the sensitivity analyses, time was divided according to changes in the most prevalent SARS-CoV-2 variant. The analyses were presented as odds ratios (ORs), incidence rates (IRRs), and 95% confidence intervals (95% CIs), as appropriate. The values were considered significant (in bold) if *p* < 0.005. Specific study periods were excluded from this analysis if no SARS-CoV-2 positive women were admitted or if cases were extremely low (sample size constraints).

## Data Availability

The anonymous data will be made available by the authors upon request and approval from the ethics committee.

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
