# Peer review of "Impact of SARS-CoV-2 Positivity on Delivery Outcomes for Pregnant Women between 2020 and 2021: A Single-Center Population-Based Analysis"

_jcm, 2023, doi:10.3390/jcm12247709_

Round 1

Reviewer 1 Report

Comments and Suggestions for Authors

The manuscript with title: ”Impact of SARS-CoV-2 positivity on delivery outcomes for pregnant women in Southern Italy between 2020 and 2021: a population-based analysis”, aims to contribute to the assessment of the impact of SARS-CoV-2 positivity on the health outcomes of pregnant women at the time of delivery and to improve our understanding to reduce vaccine hesitancy related to SARS-CoV-2 vaccination. The study behind the manuscript utilises a well known approach and the results are confirmatory. Nevertheless, the study is well made given the constraints outlined by the authors. The first part of the aim seems to be fulfilled through these well made statistical analyses. However, the second part concerning increased understanding to reduce vaccine hesitancy falls short. A couple of frequencies regarding this vaccination don’t explain much. The title itself is also problematic since this is a hospital based study and not a population based study. This problematic statement is also repeated in the method section. The authors even give an argument why this is a problematic statement in the section outlining the limitations of the study. They acknowledge that children are born elsewhere too in this region, but we are not presented with any statistics showing this distribution although this should be easy to produce.

The manuscript is basically well written but I would like to raise two points of concern. The first one concerns the disease and apparently the authors didn’t deem it necessary to consider previous episodes of covid among the pregnant women and I wonder about their argument for omitting this information, could easily be added to the manuscript. Second issue pertains to how the decision to perform a Caesarean section is made in the current setting. Italy has been reported to have a high proportion of such deliveries and while the authors want to publish a manuscript with readers from many other parts of the world this is necessary to describe and discuss both in terms of proportions and in terms of reasons behind, as well as the decision making process in general, in Italy and at this particular tertiary hospital. Good statistics without the necessary understanding of the context makes the result hard to give meaningful and necessary interpretations. It is also interesting to understand why precisely covid is singled out. What is the case for other airborne diseases like the influenza?

Another detail that needs attention is the use of the term likelihood. For instance, table 2 is presented as differences in the likelihood while this table shows ratios in the form of OR and IRR and associated CIs. In general this should be dealt with in the whole manuscript.

In the discussion (246,7) we can read that “Our results indicate that the management of infected patients during pregnancy remains suboptimal” This statement comes without any explanation or motivation, which it absolutely needs.  

Comments on the Quality of English Language

Some examples of minor problems

Introduction, line 48. Unclear what a "low level of attitude" is

Statistical section, lines 104, 105. More common expression is multiple regression.

Overall in the manuscript the use of parantheses seems unnecessary. Se for example lines 86 and 87. 

Author Response

The manuscript with title: “Impact of SARS-CoV-2 positivity on delivery outcomes for pregnant women in Southern Italy between 2020 and 2021: a population-based analysis”, aims to contribute to the assessment of the impact of SARS-CoV-2 positivity on the health outcomes of pregnant women at the time of delivery and to improve our understanding to reduce vaccine hesitancy related to SARS-CoV-2 vaccination. The study behind the manuscript utilises a well-known approach and the results are confirmatory. Nevertheless, the study is well made given the constraints outlined by the authors. The first part of the aim seems to be fulfilled through these well-made statistical analyses. However, the second part concerning increased understanding to reduce vaccine hesitancy falls short. A couple of frequencies regarding this vaccination don’t explain much. The title itself is also problematic since this is a hospital-based study and not a population-based study. This problematic statement is also repeated in the method section. The authors even give an argument why this is a problematic statement in the section outlining the limitations of the study. They acknowledge that children are born elsewhere too in this region, but we are not presented with any statistics showing this distribution although this should be easy to produce.

We would like to thank the Reviewer for the overall positive and supportive comment.

Regarding the inference on vaccine hesitancy, it was within our aim to understand why specifically women during pregnancy were less willing to accept SARS-CoV-2 vaccine. Analysing data of delivery outcomes and outlining the impact of positivity on the management of those patients might help to better inform policy makers and healthcare professionals but also the population of interest e.g. pregnant women or those who are planning to conceive. However, we agree with the Reviewer that by describing the vaccination status per se we cannot directly infer on vaccination hesitancy and we rephrased the paper accordingly to better clarify this aspect.

“Therefore, by analysing real-world data on pregnant women admitted to this academic hospital, our aim is to contribute to the assessment of the impact of SARS-CoV-2 positivity on the health outcomes of pregnant women at the time of delivery and to improve our understanding to improve the management of SARS-CoV-2 positive patients”

Although the data has been collected from a single hospital, Federico II is a tertiary referral center for pregnant woman in the Campania region and the only one which was effectively active during the first year of pandemic. Although it was not mandatory to deliver in this specific facility during the pandemic, in practice it was one the few accepting positive pregnant woman during the study period and, in particular, the only one doing it during the first year. The status of tertiary referral center was also confirmed by the total amount of deliveries included in this study, equal to 5236 for the two years, which corresponds to 6% of the total deliveries within the region (87850). The definition of population-based study even when data is retrieved from a single center which is, however, the single referral center within the area is also in line with previous literature [1]–[5].”

However, we agree with the Reviewer that this might still be considered as possible limitation. We have expanded the limitation section as follows:

“Sixth, the data was retrieved from a single center and this potentially might affect the generalization of the results. However, during pandemics “Federico II” hospital was the referral center for SARS-CoV-2 positive pregnant woman. Also, considering the total amount of deliveries analysed (6% of regional total), and the relatively large sample of positive woman included in the study might help to mitigate this potential limitation”

The manuscript is basically well written but I would like to raise two points of concern. The first one concerns the disease and apparently the authors didn’t deem it necessary to consider previous episodes of covid among the pregnant women and I wonder about their argument for omitting this information, could easily be added to the manuscript. Second issue pertains to how the decision to perform a Caesarean section is made in the current setting. Italy has been reported to have a high proportion of such deliveries and while the authors want to publish a manuscript with readers from many other parts of the world this is necessary to describe and discuss both in terms of proportions and in terms of reasons behind, as well as the decision-making process in general, in Italy and at this particular tertiary hospital. Good statistics without the necessary understanding of the context makes the result hard to give meaningful and necessary interpretations. It is also interesting to understand why precisely covid is singled out. What is the case for other airborne diseases like the influenza?

We would like to thank the Reviewer for this thoughtful comment. We agree on the importance to consider previous episodes of positivity to SARS-Cov-2, which could have contributed to explain the disease course during delivery. However, given the data architecture in the present study, it was not possible to derive this information, which has to be considered a study limitation.  Notably, during the first year of pandemic incidence of notified cases in the Campania region was relatively low, as compared with other regions. Hence, we could hypothesise that the absence of the information regarding previous SARS-Cov-2 infections might have had limited impact on our findings.

Nonetheless, we have added the following limitation in the appropriate section:

“Fourth, the data retrieved for this study was based on the content of the hospital discharge record, which did not include data on previous SARS-Cov-2 infections. Nonetheless, since very few cases of SARS-CoV-2 were notified in the first study period in the Campania region, this might have mitigated the impact of this possible bias”

As per the second comment we do acknowledge that specifically in Campania region the rate of caesarian section is among the highest in the context of both Italy and European community. Furthermore, a tertiary hospital might have even higher numbers due to the complexity of the cases admitted to the hospital. Therefore, considering the importance to provide this information for better contextualization of the study findings, we sought the opportunity to better clarify those aspects within the appropriate section adding the following line:

“Fifth, Campania region is well-known for the higher rate of registered caesarian sections. Furthermore, the complexity of cases that are treated in the University Hospital “Federico II” of Naples, a tertiary referral center, is greater than average. Hence, those two factors might have influenced the proportion of pregnant woman undergoing a caesarian section. Nonetheless, this study analysed data from a high complexity setting, which might provide information on the management of pregnant women that might be misrepresented within the general setting”

We also added further information regarding the statistics of c-section within the region in the discussion section:

Despite this rate is significantly higher than both Italian, and European rate, respectively 31.1% , and 25.7%, is consistent with the rate of 50% reported within Campania region.

Lastly, we thank the Reviewer for the opportunity to better assess the angle of this specific study: we focused on COVID-19 rather than on all airborne diseases given the very high transmission rate of SARS-CoV-2 during the pandemic and the subsequent burden on Health System. This represented an unprecedented opportunity to assess quality of delivery management in an emergency scenario”.

Another detail that needs attention is the use of the term likelihood. For instance, table 2 is presented as differences in the likelihood while this table shows ratios in the form of OR and IRR and associated CIs. In general this should be dealt with in the whole manuscript.

We would like to thank the Reviewer for this comment. We changed terminology throughout the manuscript to avoid confusion. As example:

Line 208: “Differences in the likelihood of preterm birth, caesarean section deliveries, and differences in hos-pital length of stay for delivery in SARS-CoV-2 positive women by the study period of birth or SARS-CoV-2 prevalent variant.”

In the discussion (246,7) we can read that “Our results indicate that the management of infected patients during pregnancy remains suboptimal” This statement comes without any explanation or motivation, which it absolutely needs.  

We discussed the reasons for this particular statement in the previous paragraphs. However, agree with the Reviewer that this sentence should be better clarified. Therefore, we have expanded the sentence as follows:

“Our results indicate that the management of infected patients during pregnancy might be considered as suboptimal considering the consistency of this evidence. However, further studied should be conducted considering longer study period and more granular in-formation to better assess quality of management”

Comments on the Quality of English Language

Some examples of minor problems

Introduction, line 48. Unclear what a "low level of attitude" is

We rephrased the sentence as follows “Available evidence show that pregnant women tend to have lower vaccination uptake as well as low levels of acceptancy and willingness to be vaccinated”

Statistical section, lines 104, 105. More common expression is multiple regression.

We thank the Reviewer for this comment. We defined the analysis as multivariable logistical regression, in line with what has been previously suggested [6], [7].

  • H. Katz, “Multivariable Analysis: A Primer for Readers of Medical Research,” Ann Intern Med, vol. 138, no. 8, pp. 644–650, Apr. 2003, doi: 10.7326/0003-4819-138-8-200304150-00012.Katz MH. Multivariable analysis: a primer for read- ers of medical research. Ann Intern Med. 2003;138(8):644–650.
  • E. Freedland, R. L. Reese, and B. C. Steinmeyer, “Multivariable models in biobehavioral research,” Psychosom Med, vol. 71, no. 2, pp. 205–216, Feb. 2009, doi: 10.1097/PSY.0B013E3181906E57.

Overall in the manuscript the use of parentheses seems unnecessary. Se for example lines 86 and 87. 

We reduced the use of parentheses where possible.

References

  1. Sun et al., “A Population-Based Systematic Clinical Analysis With a Single-Center Case Series of Patients With Pulmonary Large Cell Neuroendocrine Carcinoma,” Front Endocrinol (Lausanne), vol. 12, p. 759915, Dec. 2021, doi: 10.3389/FENDO.2021.759915/BIBTEX.
  2. A. Barzenje, A. Kolstad, W. Ghanima, and H. Holte, “Long-term outcome of patients with solitary plasmacytoma treated with radiotherapy: A population-based, single-center study with median follow-up of 13.7 years,” Hematol Oncol, vol. 36, no. 1, pp. 217–223, Feb. 2018, doi: 10.1002/HON.2415.
  3. Salvi et al., “A seasonal periodicity in relapses of multiple sclerosis? A single-center, population-based, preliminary study conducted in Bologna, Italy,” BMC Neurol, vol. 10, no. 1, pp. 1–6, Nov. 2010, doi: 10.1186/1471-2377-10-105/TABLES/1.
  4. Floodeen, R. Lindgren, and P. Matthiessen, “When are defunctioning stomas in rectal cancer surgery really reversed? result s from a population-based single center experience,” Scandinavian Journal of Surgery, vol. 102, no. 4, pp. 246–250, Sep. 2013, doi: 10.1177/1457496913489086/ASSET/IMAGES/LARGE/10.1177_1457496913489086-FIG1.JPEG.
  5. Greenbaum et al., “Cesarean Delivery and Childhood Malignancies: A Single-Center, Population-Based Cohort Study,” Journal of Pediatrics, vol. 197, pp. 292-296.e3, Jun. 2018, doi: 10.1016/j.jpeds.2017.12.049.

Reviewer 2 Report

Comments and Suggestions for Authors

Thank you for giving me a chance to review this study. It is a good Population based analysis that present a Impact of SARS-CoV-2 positivity on delivery outcomes for  pregnant women in Southern Italy between 2020 and 2021: a population-based analysis

Dear author,

Comments

1.     Abstract: Include more results

2.     For Referencing you have used () brackets please check the guidelines it should be like this [].

Introduction: the following statement in the introduction does not correspond to the Reference no 1

The COVID-19 pandemic started in January 2020 has served as a benchmark for 35 health systems particularly for emergency departments (1).

The SARS-CoV-2 virus can 36 cause respiratory infections, including severe pneumonia, and in a small percentage of 37 cases, it can lead to death (2).

Pregnancy can influence the response to SARS-CoV-2 due 38 to physiological changes that affect the cardiorespiratory and immune systems (3).

1.      Xie F, Zhou J, Lee JW, Tan M, Li S, Rajnthern LS /O, et al. Benchmarking emergency department prediction models with machine 301 learning and public electronic health records. Scientific Data 2022 9:1. 2022;9(1): 1–12. https://doi.org/10.1038/s41597-022-01782-302 9.

2.      Schwartz DA, Hyg M. An Analysis of 38 Pregnant Women With COVID-19, Their Newborn Infants, and Maternal-Fetal Trans-304 mission of SARS-CoV-2 Maternal Coronavirus Infections and Pregnancy Outcomes.

3.      Wastnedge EAN, Reynolds RM, van Boeckel SR, Stock SJ, Denison FC, Maybin JA, et al. Pregnancy and COVID-19. Physiological 306 reviews. 2021;101(1): 303–318. https://doi.org/10.1152/PHYSREV.00024.2020. 

Reference 4. Not correlated with the mentioned Reference it  is contrast.  Early findings indicated that pregnancy was not strongly associated with an in-40 creased risk of SARS-CoV-2 infection or symptomatic COVID-19(4)

4.      Allotey J, Stallings E, Bonet M, Yap M, Chatterjee S, Kew T, et al. Clinical manifestations, risk factors, and maternal and perinatal 308 outcomes of coronavirus disease 2019 in pregnancy: living systematic review and meta-analysis. BMJ. 2020;370. 309 https://doi.org/10.1136/BMJ.M3320. 310

3.     Method: Authors did not mention the exclusion criteria

4.     Authors mentioned 55 vaginal delivery and 94 caesarean among the study population.

Better provide explanation

5.     Authors did not mention the number of pregnancy of the women included in the study.

6.     Line 241 measures, (29), (30) (31), please correct it as measures (29), (30) (31).

7.     Statistics-Why did you choose sensitivity analysis and stratified analysis; explain better

8.     Please highlight how this study adds to the current available knowledge.

9.     Please include future recommendations for this study

10.  Kindly add the strength and limitations of the study…

11.  Discussion: Very less References: Please include more literature relevant to your study.

Comments on the Quality of English Language

Minor editing of english language is required

Author Response

Thank you for giving me a chance to review this study. It is a good Population based analysis that present a “Impact of SARS-CoV-2 positivity on delivery outcomes for  pregnant women in Southern Italy between 2020 and 2021: a population-based analysis

Dear Reviewer,

We would like to thank you for the thoughtful and overall positive comment, we sought to answer point by point to facilitate your work.

  1. Abstract:Include more results

  1. For Referencing you have used () brackets please check the guidelines it should be like this [].

Thank for the suggestions, to address the first comment we added further relevant results in the abstract. In particular, the following line was added “The Obstetric Comorbidity Index was higher than 0 in 41% of the sample” and “it was confirmed adjusting for SARS-CoV-2 variant.”

Then to address the second comment, “()” brackets have been substituted with “[]” throughout the paper.

Introduction: the following statement in the introduction does not correspond to the Reference no 1

The COVID-19 pandemic started in January 2020 has served as a benchmark for 35 health systems particularly for emergency departments (1).

The SARS-CoV-2 virus can 36 cause respiratory infections, including severe pneumonia, and in a small percentage of 37 cases, it can lead to death (2).

Pregnancy can influence the response to SARS-CoV-2 due 38 to physiological changes that affect the cardiorespiratory and immune systems (3).

  1. Xie F, Zhou J, Lee JW, Tan M, Li S, Rajnthern LS /O, et al. Benchmarking emergency department prediction models with machine 301 learning and public electronic health records. Scientific Data 2022 9:1. 2022;9(1): 1–12. https://doi.org/10.1038/s41597-022-01782-302 9.
  2. Schwartz DA, Hyg M. An Analysis of 38 Pregnant Women With COVID-19, Their Newborn Infants, and Maternal-Fetal Trans-304 mission of SARS-CoV-2 Maternal Coronavirus Infections and Pregnancy Outcomes.
  3. Wastnedge EAN, Reynolds RM, van Boeckel SR, Stock SJ, Denison FC, Maybin JA, et al. Pregnancy and COVID-19. Physiological 306 reviews. 2021;101(1): 303–318. https://doi.org/10.1152/PHYSREV.00024.2020. 

Reference 4. Not correlated with the mentioned Reference it  is contrast.  Early findings indicated that pregnancy was not strongly associated with an in-40 creased risk of SARS-CoV-2 infection or symptomatic COVID-19(4)

  1. Allotey J, Stallings E, Bonet M, Yap M, Chatterjee S, Kew T, et al. Clinical manifestations, risk factors, and maternal and perinatal 308 outcomes of coronavirus disease 2019 in pregnancy: living systematic review and meta-analysis. BMJ. 2020;370. 309 https://doi.org/10.1136/BMJ.M3320. 310

We would like to thank the reviewer for this comment. We are deeply sorry for these mistakes which occurred during the manuscript formatting according to publication guidelines. We have fixed the issue and now all references are correctly cited. We also improved our bibliography with more references. For the specific comment on the “4”th citation, we corrected the line with: “Early findings indicated that pregnancy was associated with a reduced risk of manifesting low or mild symptomatic COVID-19”.

  1. Method: Authors did not mention the exclusion criteria

 We would like to thank the reviewer for those annotations. We better clarified the criteria that were taken in account in defining the population for this study. We have added to the paper: “All records of women who gave birth at the University Hospital "Federico II" in Naples between January 2020 and December 2021 were included in the analysis as they did not contain missing data for the study variables.”

  1. Authors mentioned 55 vaginal delivery and 94 caesarean among the study population.

Better provide explanation

 Thank for the comment. We invite the Reviewer to read the answer to Reviewer 1  for detailed explanation. In summary, this data corresponded to the total of SARS-CoV-2 positive woman delivering at the Federico II hospital during 2020 (149 deliveries). The percentage of caesarean section is higher than expected for two main reasons. First, as a regional tertiary referral centre, the case mix of this hospital is higher than the average within the regional hospitals. Second, caesarean section rate in Campania region is significantly higher than average Italian and European rate. 

This aspect has been delt with within the paper, for example at line 221-223 you may find the newly added line “Despite this rate is significantly higher than both Italian, and European rate, respectively 31.1% , and 25.7%, is consistent with the rate of 50% reported within Campania region [8], [9]” which in our opinion help to better clarify this aspect.

  1. Authors did not mention the number of pregnancy of the women included in the study.

We would like to thank the reviewer for the comment. To provide this important information we have added the following line: “We obtained data for 5,236 pregnant women and deliveries that occurred between January 2020 and December 2021.”

  1. Line 241 measures, (29), (30) (31), please correct it as measures (29), (30) (31).

Corrected.

  1. Statistics-Why did you choose sensitivity analysis and stratified analysis; explain better

We would like to thank the reviewer for this thoughtful comment.

The main analysis considered equally spaced time periods to better assess whether management of pregnant women differed between those who were positive and negative to SARS-CoV-2 at time of labour. Whilst one of the sensitivity analyses divided the time period considering the change in predominant variants over time to focus on whether this factor might explain differences in study period, considering that there is evidence supporting the change of nature and extent of clinical symptoms by variant type [10]–[12], the second sensitivity analysis only focused on pregnant women positive to SARS-CoV-2 at time of labour to better assess change in management of this specific population over time.

We have expanded the specific section as follows:

“When we restricted our analyses to individuals who tested positive for SARS-CoV-2 at the time of labour, which was conducted to better assess change in management of this specific population over time,  we found that, compared with the October-December 2020 study period, during the July-September 2021 study period, the likelihood of a caesarean section decreased by 61% (OR 0.39, 95% CI 0.19-0.79). No differences were observed in the likelihood of preterm delivery or the incidence rate of length of stay (Table 2).”

And “When we stratified our analyses based on the change in the most prevalent SARS-CoV-2 variant over time to focus on whether this factor might explain differences, rather than using three-month study periods, no differences were observed in the likelihood of preterm birth”

  1. Please highlight how this study adds to the current available knowledge.

We would like to thank the reviewer for helping us better clarify the contribution of this specific study to the existing scientific knowledge. Although availability of data and papers of the  pandemic era represent now a large body of evidence, evidence from some specific settings is still falling short of a complete coverage. The University Hospital “Federico II” of Naples is a tertiary referral center for delivery and has admitted almost the entire population of SARS-CoV-2 positive women at time of labour and has recorded the highest number of births within the region during the analysed study period. This study is based on the full dataset of the most critical part of the pandemic. We also sought to add the following line to the conclusion part to better assess this aspect:

“Our findings provide data from a high complexity setting that might help to better assess the impact of SARS-CoV-2 in managing patients during delivery and improve generalizability of existing evidence.”

  1. Please include future recommendations for this study

Thank for the suggestion, we have added the following line to the conclusion section:

 “Further studies focusing on the comparison of delivery outcomes between settings that differ in clinical case complexity and activity volume might help to better assess the impact of airborne communicable diseases during delivery as well as identify drivers of improved management.”

  1. Kindly add the strength and limitations of the study…

We would like to thank the reviewer for this comment.  We have now better highlighted the strengths and limitations of the study in a dedicated paragraph.

  1. Discussion: Very less References: Please include more literature relevant to your study.

We have now extended the list of references we have cited.

[3]       A. Sharma, S. Tiwari, M. K. Deb, and J. L. Marty, “Severe acute respiratory syndrome coronavirus-2 (SARS-CoV-2): a global pandemic and treatment strategies,” Int J Antimicrob Agents, vol. 56, no. 2, p. 106054, Aug. 2020, doi: 10.1016/J.IJANTIMICAG.2020.106054.

[4]       B. Hu, H. Guo, P. Zhou, and Z. L. Shi, “Characteristics of SARS-CoV-2 and COVID-19,” Nature Reviews Mi-crobiology 2020 19:3, vol. 19, no. 3, pp. 141–154, Oct. 2020, doi: 10.1038/s41579-020-00459-7.

[26]     “Caesarean section rates continue to rise, amid growing inequalities in access.” Accessed: Dec. 02, 2023. [Online]. Available: https://www.who.int/news/item/16-06-2021-caesarean-section-rates-continue-to-rise-amid-growing-inequalities-in-access

[27]     “Certificato di assistenza al parto (CeDAP). Analisi dell’evento nascita - Anno 2020.” Accessed: Dec. 02, 2023. [Online]. Available: https://www.salute.gov.it/portale/documentazione/p6_2_2_1.jsp?lingua=italiano&id=3149

Round 2

Reviewer 1 Report

Comments and Suggestions for Authors

I want to thank the authors of the manuscript “Impact of SARS-CoV-2 positivity on delivery outcomes for pregnant women in Southern Italy between 2020 and 2021: a population-based analysis” for their efforts to improve the text and reply to the concerns I have presented. They have done a good job and will present my remaining issues below.

1.       The suggested change in aim is a clear improvement. The authors state the new aim as: “Therefore, by analysing real-world data on pregnant women admitted to this academic hospital, our aim is to contribute to the assessment of the impact of SARS-CoV-2 positivity on the health outcomes of pregnant women at the time of delivery and to improve our understanding to improve the management of SARS-CoV-2 positive patients” An English check should be employed to also avoid repeating words and all will be fine.

2.       Concerning my critique regarding the use of the statement “a population-based analysis” which exist in the title and elsewhere too, the authors says this is fine to use and then tries to corroborate this statement by providing 5 literature references. I thank the authors for these references since they further help me to clarify my point. In every title in this list the authors mention “Single-Center” and this is also absolutely necessary in this current manuscript, both in text and title. It is important to avoid confusion concerning the scope of the manuscript.

3.       In my previous comments to the authors I wrote that “Another detail that needs attention is the use of the term likelihood. For instance, table 2 is presented as differences in the likelihood while this table shows ratios in the form of OR and IRR and associated CIs. In general this should be dealt with in the whole manuscript.” Clearly my comment was not properly understood. In the latest manuscript I count 20 instances of this word. Since this manuscript relies on advanced statistics I find it prudent to be careful with the use of terms that are common in statistics and have a special meaning. My sense is that the authors use the likelihood like they use the word probability or risk. Se for example the heading of table 2. The table does not contain any measures which are associated with the term likelihood in a statistical sense and therefore the heading must be changed to reflect the actual content of the table which consists of different ratios i.e. OR and IRR and associated CIs and not any likelihoods. I still claim this has to be remedied throughout the whole manuscript, for the sake of clarity.

4.       Last issue I want to address pertains to how the decision to perform a Caesarean section is made in the current setting. The authors have acknowledged that Italy differs from other countries  but also that different parts of Italy are different, but I still didn’t see any attempt to explain how these decisions are made in Italy, in this region and in particular at this tertiary hospital. Is it possible for women to require this procedure, is it a decision made by doctors alone or is it a combination.  The question is then how do these decision procedures affect the results and conclusions made in this manuscript.

Comments on the Quality of English Language

Comments regarding language is found in the previous section

Author Response

I want to thank the authors of the manuscript “Impact of SARS-CoV-2 positivity on delivery outcomes for pregnant women in Southern Italy between 2020 and 2021: a population-based analysis” for their efforts to improve the text and reply to the concerns I have presented. They have done a good job and will present my remaining issues below.

We would like to thank the Reviewer for the positive comment. We will now answer point by point to the comments raised.

  1. The suggested change in aim is a clear improvement. The authors state the new aim as: “Therefore, by analysing real-world data on pregnant women admitted to this academic hospital, our aim is to contribute to the assessment of the impact of SARS-CoV-2 positivity on the health outcomes of pregnant women at the time of delivery and to improve our understanding to improve the management of SARS-CoV-2 positive patients” An English check should be employed to also avoid repeating words and all will be fine.

We thank the Reviewer again for the comment raised; we provided proofreading this specific statement and changed as it follows:

“Therefore, by analyzing real-world data on pregnant women admitted to this academic hospital, we aim to contribute to the assessment of the impact of SARS-CoV-2 positivity on the health outcomes of pregnant women at the time of delivery and subsequently improve the management of SARS-CoV-2 positive patients.”

  1. Concerning my critique regarding the use of the statement “a population-based analysis” which exist in the title and elsewhere too, the authors says this is fine to use and then tries to corroborate this statement by providing 5 literature references. I thank the authors for these references since they further help me to clarify my point. In every title in this list the authors mention “Single-Center” and this is also absolutely necessary in this current manuscript, both in text and title. It is important to avoid confusion concerning the scope of the manuscript.

We would like to thank the Reviewer for further clarify this aspect. We have now specified throughout the manuscript and the title that the present study was conducted within a “Single-Center”_

E.g.

The title has been changed in “Impact of SARS-CoV-2 positivity on delivery outcomes for pregnant women between 2020 and 2021: a single-center population-based analysis”.

In the discussion session: “In this retrospective single-center population-based observational study, which included data on the entire population of women who gave birth at the University Hospital 'Federico II' of Naples,”

  1. In my previous comments to the authors I wrote that “Another detail that needs attention is the use of the term likelihood. For instance, table 2 is presented as differences in the likelihood while this table shows ratios in the form of OR and IRR and associated CIs. In general this should be dealt with in the whole manuscript.” Clearly my comment was not properly understood. In the latest manuscript I count 20 instances of this word. Since this manuscript relies on advanced statistics I find it prudent to be careful with the use of terms that are common in statistics and have a special meaning. My sense is that the authors use the likelihood like they use the word probability or risk. Se for example the heading of table 2. The table does not contain any measures which are associated with the term likelihood in a statistical sense and therefore the heading must be changed to reflect the actual content of the table which consists of different ratios i.e. OR and IRR and associated CIs and not any likelihoods. I still claim this has to be remedied throughout the whole manuscript, for the sake of clarity.

We would like to thank the Reviewer for this comment, we provided to change the statistical terminology used throughout the paper. As example:

“If compared with negative women, positive ones had an 80% increased odds of caesarean section, and it was confirmed adjusting for SARS-CoV-2 variant. No significant differences were found in preterm birth risk.”

  1. Last issue I want to address pertains to how the decision to perform a Caesarean section is made in the current setting. The authors have acknowledged that Italy differs from other countries but also that different parts of Italy are different, but I still didn’t see any attempt to explain how these decisions are made in Italy, in this region and in particular at this tertiary hospital. Is it possible for women to require this procedure, is it a decision made by doctors alone or is it a combination. The question is then how do these decision procedures affect the results and conclusions made in this manuscript.

We would like to thank the Reviewer for this insightful comment and raising an important aspect of our study. We agree on the importance of increase clarity on the decision-making process regarding Caesarean sections in the specific setting.

The decision to perform a caesarian section, although it has to be accepted by the patient, stands as a prerogative of the gynecologist, which applies to every setting. When focusing on pregnant women positive to SARS-CoV-2, evidence suggests an increase in caesarian sections worldwide, despite the guidelines suggesting differently. To further clarify this aspect, we included the following line in the introduction within the section addressing the possible influential factors for delivery outcomes:

The most common pregnancy-related complications associated with SARS-CoV-2 infection appear to be premature rupture of membranes (PROM) (5%), foetal distress (14%), and postpartum fever (8%) [14], [15], with a higher risk of requiring caesarean section and experiencing preterm delivery compared to uninfected pregnant women [16]. “Evidence suggests that, despite what recommended in clinical guidelines, the positivity to SARS-CoV-2 might led the caregiver to prefer a caesarean delivery (https://doi.org/10.1016/j.cmi.2020.10.007)”. Some studies have reported a reduced risk of preterm birth during the pandemic period, as compared to the pre-pandemic years [17], [18].

Furthermore, to better frame the setting in which present study has been conducted we modified the following statement:

“On average, 47% of positive women have had caesarean section. This percentage is significantly higher than both Italian, and European rate, 31%, and 26%, respectively, while is consistent with regional statistics.”

Italy and Campania region were reported for being above average for caesarean section long before pandemic started. Reason behind this trend is not framed accurately in literature and furthermore they are beyond our aims. Nonetheless we do agree with the reviewer that a context might help the readers from any part of the globe to better understand the data presented. Therefore, we added also these lines:

" In Italy, several factors impact the increased rate of cesarean sections as the perception of professional risk (https://doi.org/10.1787/9789264266414-en), low adherence to recommendations (https://doi.org/10.1186/1471-2393-12-54), and the pursuit of profit in private facilities (https://doi.org/10.1186/s12884-020-03462-1).”

And also “Nonetheless, this trend was reported long before the pandemic started (https://doi.org/10.1186/s12884-020-03462-1) and therefore this result needs to be further investigated.”

We would also like to point out that although the caesarian section rate is higher than average in the Campania region, our study design might have mitigated this possible source of bias as this higher rate of caesarian sections refers to the entire study population, regardless of the result of the SARS-CoV-2 test.